# Analysis of the Impact of Remanufacturing Process Innovation on Closed-Loop Supply Chain from the Perspective of Government Subsidy

**Kailing Liu, Quanxi Li *** and **Haowei Zhang**

School of Business and Management, Jilin University, Changchun 130022, China
* Correspondence: quanxili.jlu@gmail.com

**Abstract:** Efficient and low-cost remanufacturing can be achieved through process innovation. Consequently, the question of whether government subsidies for remanufacturing process innovation will stimulate improvement in this area and thus affect the closed-loop supply chain is worth discussing. To answer this question, we establish a closed-loop supply chain model consisting of a manufacturer and a retailer, taking into account both remanufacturing process innovation and government subsidy. This is done in order to explore the impact of remanufacturing process innovation on the closed-loop supply chain from the perspective of government subsidies. Moreover, the government subsidizes the manufacturer according to the improvement of remanufacturing process innovation levels. Specifically, we analyze the optimal decisions and the social welfare in two models—the benchmark model without government subsidy, and the government subsidy model. Our main findings are threefold. The optimal decisions of the two models vary with the proportion of new products' production cost and remanufactured products' production cost. The government subsidy for process innovation does not necessarily improve the profits of the manufacturer, the retailer, and the supply chain system. Moreover, there is a threshold; the government subsidy can hurt the retailer's profits, and the retailer has no motivation to participate in the sale of new products when the government subsidy is below that threshold. The government subsidy for process innovation does improve overall social welfare and has a lesser environmental impact. The conclusions are also verified by numerical analysis.

**Keywords:** closed-loop supply chain; process innovation; government subsidy; remanufacturing



## 1. Introduction

Due to convoluted environmental laws and business economic situations, more and more people are focusing their attention on the operation and development of the closed-loop supply chain [1]. The closed-loop supply chain aims to close the flow of materials, reduce pollution emissions and residual waste, and provide services to customers at a lower cost. It includes traditional forward and recovery logistics [2]. Many scholars have conducted relevant research on closed-loop supply chains, such as pricing decision-making, optimal recovery rate, channel structure, and so on [3–5]. However, as the life cycle of mobile phones and computers is shortened and the replacement rate is accelerated, the waste of resources and the shortage of raw materials has become a significant problem in today's society [6]. Moreover, consumers' behavior will cause an increase in waste and natural resource destruction when there is no closed-loop supply chain [7]. According to Menad et al. [8], electrical and electronic waste mass is increasing with technological growth. For instance, Kaya [9] quoted from the www.powershow.com (accessed on 7 January 2010). website that the waste recycling rate in the U.S. in 2010 was 40% computers (423,000 tons), 33% monitors (595,000 tons), 11% mobile phones (19,500 tons), 10% keyboards and mice (67,800 tone), and 17% TVs (1,045,000 tons); also 40,000 mobile phones are discarded in the U.S. every day. If these recycled electronic products are remanufactured, they can not only reduce costs, reduce resource and energy consumption, and reduce environmental pollution, but also

achieve maximum economic benefits with minimum investment. Therefore, closed-loop supply chain remanufacturing becomes an urgent matter to be solved.

In a situation where resources and energy will become increasingly scarce, remanufacturing has apparent advantages. According to the relevant investigation, compared with the new products, remanufactured products can save costs by 50%, save energy by 60%, save materials by 70%, and save air pollution emissions by more than 80% [10]. Remanufacturing has benefits, but there are certain thresholds and restrictions, and not every enterprise can enter at will. As is known to all, the product remanufacturing process generally includes the following steps: product cleaning, target object disassembly, cleaning, testing, remanufacturing parts classification, remanufacturing technology selection, remanufacturing, and inspection. However, remanufacturing is very expensive no matter the processing equipment, testing equipment, or cleaning equipment that is needed. Process innovation can reduce production costs and save time without compromising the quality of products or services. Henry Ford, for example, made the assembly of his cars less complicated through process innovation, with the duration reduced from 12 h to 90 min. The reduction in production costs was reflected in the Model T in 1908, which came at a decreased price. This price increased the number of customers (data from https://checkify.com/blog/process-innovation/) (accessed on 14 March 2020). Therefore, we also need to consider whether efficient and low-cost remanufacturing can be achieved through process innovation.

Remanufacturing process innovation has been regarded as the essential means to seek sustainable development and improve the competitiveness of enterprises [11,12]. Taking Apple's progress report [13], for example, Daisy, the traditional disassembling robot, can disassemble an iPhone into various parts, whereas Dave, the latest disassembling robot, can further disassemble the touch engine to recover magnets containing rare earth elements. The improved disassembly process and technology saves time and manpower for Apple and avoids waste, thus reducing the cost of remanufacturing. Meanwhile, it also actively responds to the national energy conservation and emission reduction policy by reducing carbon emissions caused by product use and maximizing social welfare through process innovation and technological innovation [14]. Successful process innovation can enhance competitive advantages and sustainability by increasing output, reducing life-cycle costs, reducing environmental impact, and improving production efficiency [15]. However, as a means of saving resources and avoiding waste, remanufacturing is discouraged by many enterprises because of high remanufacturing costs. If the remanufactured product is not cost-effective, the enterprise will have no incentive to carry out further remanufacturing process innovation. Therefore, in order to increase enterprises' remanufacturing profits and respond to the call for sustainable development strategy, it is necessary to reduce remanufacturing costs through process innovation. In addition, the reduction of remanufacturing costs can enable enterprises to increase consumers' demand for remanufactured products by adopting appropriate pricing strategies (such as low price strategy or price reduction strategy) [16]. Although remanufacturing process innovation can reduce the remanufacturing costs of enterprises, it will also bring innovation costs to enterprises. When the innovation costs are greater than the remanufacturing costs, the manufacturer will have no motivation for process innovation. It is possible that governments can adopt appropriate incentive policies, such as subsidies. In that case, governments can promote the development of remanufacturing and support the maximization of social welfare and environmental protection.

However, the current incentive measures of the Chinese government are relatively simple, that is, "simple and rough" direct subsidies or different subsidy levels. Few studies subsidize remanufacturing based on the level of improvement in remanufacturing process innovation. Therefore, to fill this gap, this paper studies a closed-loop supply chain with remanufacturing, whereby the remanufacturing costs could be reduced through remanufacturing process innovation. Unlike traditional direct government subsidies, the subsidy is introduced based on the remanufacturing process innovation level discussed

in this paper. In other words, the higher the manufacturer's remanufacturing process innovation level is, the higher the manufacturer's government subsidy will be.

In this paper, these models are used to answer the following questions:

(1) What are the equilibrium decisions in the benchmark model without government subsidy and the government subsidy model?
(2) Can the manufacturer benefit from the government subsidy as the level of remanufacturing process innovation increases?
(3) Under what conditions will the manufacturer choose to improve its remanufacturing process innovation level?
(4) Can government subsidies improve overall social welfare? What is the impact on the environment?

To answer these questions, we establish two models: the benchmark models with remanufacturing process innovation but no government subsidies, and the model with remanufacturing process innovation and the government subsidy. We obtain the equilibrium decisions of the two models by backward induction and investigate the changes in equilibrium decisions with consumers' acceptance of remanufactured products and remanufacturing process innovation cost coefficient. We also study the influence of government subsidies on the whole of social welfare through manufacturer profit, consumer surplus, and environmental impact. We also verify the outcomes by numerical analysis.

The content construction of this paper is as follows: Section 2 reviews the previous literature. In Section 3, we provide the model formulation and relevant model assumptions. The model analysis and solutions are provided in Section 4. In Section 5, the results of models are compared and analyzed, and some conclusions are obtained. In Section 6, the numerical analysis further verifies the optimal solutions and related conclusions. We conducted comparative analyses of consumer surplus, environmental impact, and social welfare in Section 7. The conclusions and limitations of our paper are presented in Section 8.

## 2. Literature Review

Our research deals with several significant literature streams: closed-loop supply chain and remanufacturing, process innovation, and government subsidy. Next, we review the relevant literature and clarify our contributions.

The first related research stream is a closed-loop supply chain and remanufacturing. There is already much literature on the closed-loop supply chain, including pricing decisions, retailer competition, supply chain collaboration, optimal recycling strategy, and so on. Based on the game theory, F. Zhang et al. [17] analyze the optimal recovery rate, price decisions and system efficiency of the closed-loop supply chain under the competition between online retailers and traditional retailers. X.-X. Zheng et al. [18] establish a three-echelon closed-loop supply chain that includes a manufacturer, a distributor and a fairness concerns retailer. Furthermore, the study adopts cooperative game theory to coordinate the supply chain systems. Based on Jafari et al. [19], who developed a dual-channel closed-loop supply chain consisting of a collector, a recycler, and a manufacturer, Giri et al. [20] extend their models with a backup supplier considering the uncertainty of the collection of used products. Z. Liu et al. [21] establish a two-echelon closed-loop supply chain to explore the effects of product design on its operations. Moreover, the study finds that the adjustment of product design can curb loss if profitability suffers. Hong et al. [22] built three collection models: a retailer collection model, a cooperative collection model (i.e., the manufacturer cooperates with a third-party firm), and a subcontracted collection model (i.e., the manufacturer subcontracted a third-party firm). This paper aims to evaluate the advantages and disadvantages of the three models. Gao et al. [3] establish three decentralized models to explore optimal pricing decision-making. Under the situation of remanufacturing costs being disrupted, H. Wu et al. [23] research the production and coordination decisions in a closed-loop supply chain consisting of a manufacturer and two competing retailers. Xie et al. [24] establish a dual-channel closed-loop supply chain, in which the products are sold online and offline in the forward channel, and the used products are recycled in

the offline channel to solve the problem of forwarding channel conflicts and improve the quality of reverse-channel recycling products. All the above works focus on exploring the optimal pricing strategy (F. Zhang et al. [17], Gao et al. [3]), optimal recovery strategy (Giri et al. [20], H. Wu et al. [23]) or supply chain coordination (X.-X. Zheng et al. [18], Z. Liu et al. [21], H. Wu et al. [23]) in the closed-loop supply chain.

There is also some literature that takes remanufacturing into consideration in the closed-loop supply chain. According to a stylized model of endogenous product quality improvement and remanufacturing, G. D. Li et al. [25] find that the main driver of the contradicting results is the change in manufacturing costs caused by improving product quality. X. Wu et al. [26] develop a game-theoretical model to investigate the role of nondiscriminatory uniform pricing policy and buyer-specific pricing policy in the closed-loop supply chain. In addition, research shows that third-party remanufacturing can lead to a triple win for the supplier, the original equipment manufacturer, and the third-party remanufacturer regardless of the pricing policy. Giri et al. [27] investigate a closed-loop supply chain with a forwarding dual channel. The manufacturer sells products through traditional retail channels, e-tail (Internet) channels, and a reverse dual channel. The manufacturer collects used products for remanufacturing through traditional third-party logistics and e-tail channels.

The second related research stream is process innovation in the supply chain. Ayhan et al. [28] propose a new performance index for monitoring and measuring manufacturing process innovation—process innovation degree. Adner et al. [29] develop a formal computer simulation model to examine the dynamics of product and process innovation. Dobson et al. [30] finds that process innovation can increase operational efficiency through a step-change improvement in resource utilization and that waste reduction can help boost manufacturing profitability and offer broader social and environmental benefits. In addition, the involvement of the government can stimulate process innovation to support lean supply chains and sustainability. Reimann et al. [31] mainly study the relationship between remanufacturing and reducing variable remanufacturing costs through process innovation. S. D. Li [32] establishes a dynamic control model for product and process innovation of multi-product monopolists, which takes into account knowledge accumulated through practical learning. This paper aims to study the optimal decision-making behavior of multi-product monopolists investing in product and process innovation and accumulating knowledge through practical learning under monopolistic and social planner optimal conditions. Chenavaz [33] models dynamic pricing, product and process investment policies in optimal control settings. Wang et al. [34] adopt a dynamic game with knowledge accumulation to investigate the optimal research-and-development portfolio of a single-product monopolist investment in product and process innovations of a South-country firm. Based on game theory, Genc et al. [35] explore the impact of some innovation-led lean programs on the sustainability of manufacturers and suppliers in the setting of a closed-loop supply chain. In order to evaluate the effect of business process innovation on the relationship between social quality management and supply chain performance, Schniederjans [36] has verified the positive association between them by hierarchical moderated regression analyses. B. Liu et al. [37] build a dynamic supply chain model consisting of a supplier and a manufacturer. Moreover, the supplier implements green process innovation referring to industry 4.0 technologies. This paper shows that the overall benefit of environmental cooperation exists in a profit-Pareto-improving region in green process innovation. In order to promote the understanding of the relationship between the supply chain and the innovation process, Zimmermann et al. [38] analyze 94 papers from 37 journals and discuss their major contributions by means of a systematic literature review. Through structural equation analysis of data from 374 manufacturing enterprises, Cherrafi et al. [39] obtain the relationship between lean, green process-innovation practices and green supply chain performance. Chang et al. [40] establish a dynamic supply chain including a manufacturer and two retailers to investigate how the retailer with process innovation adapts control strategies to acquire an advantage over the competition in the face of competition from

the manufacturer and another retailer. In order to achieve more efficient, lean, and sustainable operations, Ni et al. [41] establish a dynamic supply chain to explore the pricing and sales strategies of the manufacturer with different innovation efficiencies under the condition of information asymmetry. All the above literature reflects the benefits of process innovation in remanufacturing (Reimann et al. [31], etc.), product production (S. D. Li [32], Chenavaz [33], Wang et al. [34], etc.), environmental sustainability (Dobson et al. [30], etc.), and so on. At the same time, it also provides references for exploring the influence of remanufacturing process innovation on the closed-loop supply chain.

The third related research stream is government subsidy. Although remanufacturing process innovation can reduce remanufacturing costs, we cannot ignore process innovation costs. Moreover, government subsidies can have a positive impact on manufacturers to a certain extent. Z. Li et al. [42] investigate the decision-making changes before and after government subsidy in manufacturer leadership, remanufacturer leadership, and no leadership models. The study also takes social welfare into consideration. Considering the remanufacturing government subsidy and the market environment, Feng et al. [43] analyze the full remanufacturing and partial remanufacturing duopoly game models by Stackelberg and Cournot. Mondal et al. [44] aim to investigate how the corporate social responsibility (CSR) of retailers can affect supply chain members' optimal decisions by building an integrated model and three manufacturer-led decentralized models. Based on the centralized Nash game, and manufacturer-led Stackelberg game, Mondal et al. [45] built a two-level, closed-loop supply chain consisting of two competing manufacturers and a common retailer to market substitutable products under government sponsorship. This paper aims to explore the manufacturer, the optimal results of government intervention, and the improvement of supply chain efficiency. He et al. [46] establish a dual-channel closed-loop supply chain to explore the channel structure, pricing decisions, and government's subsidy policy when new products compete with remanufactured products. The study finds that an appropriate government subsidy level can encourage the manufacturer to adapt the ideal channel structure. Huang et al. [47] establish CLSC models of three dual-channel recycling modes to investigate the influence of government subsidies on channel members' pricing decisions and recycling mode selection.

This paper is similar to Reimann et al. [31] and Chai et al. [48], which also consider remanufacturing process innovation in the closed-loop supply chain. However, the difference is that the government subsidy for remanufacturing process innovation is considered in this paper, and the subsidy intensity that manufacturers can obtain is determined by the level of process innovation. Previous studies may only provide subsidies for remanufacturing in the closed-loop supply chain without considering process innovation (see Feng et al. [43]), and this paper makes up this gap. In this paper, the government subsidies are different from the previous subsidies for product sales (the more the sales are, the more the subsidies are); that is, the government subsidizes the enterprise based on the improvement of the innovation level of the remanufacturing process. In the benchmark model, the enterprise has implemented remanufacturing process innovation, but its innovation level is not high without government support. When the enterprise knows that the government will subsidize remanufacturing process innovation, it will improve the remanufacturing process innovation level in order to obtain higher subsidies. Research finds that government subsidies can stimulate the manufacturer to improve process innovation level, but the improvement degree of process innovation level is affected by the level of government subsidies. The optimal decisions are influenced by the consumers' acceptance of remanufactured products and the process innovation coefficient. In addition, the overall social welfare can be improved when the government participates in the closed-loop supply chain. Table 1 shows the comparison between the previous relevant literature and our paper.

**Table 1.** The related literature.

| Literature | Process Innovation | Remanufacturing | Closed-Loop Supply Chain | Government Subsidy |
|---|---|---|---|---|
| H.T. Chen et al. [16] (2021) | √ | √ | √ | × |
| Ayhan et al. [28] (2013) | √ | × | × | × |
| Adner et al. [29] (2001) | √ | × | × | × |
| Reimann et al. [31] (2019) | √ | √ | √ | × |
| S.D. Li [32] (2018) | √ | × | × | × |
| Chenavaz [33] (2012) | √ | × | × | × |
| Genc et al. [35] (2020) | √ | × | √ | × |
| B. Liu et al. [37] (2019) | √ | × | × | × |
| Cherrafi et al. [39] (2018) | √ | × | × | × |
| Chang et al. [40] (2021) | √ | × | × | × |
| Z. Li et al. [42] (2019) | × | √ | × | √ |
| Feng et al. [43] (2021) | × | √ | √ | √ |
| Mondal et al. [45] (2021) | × | × | √ | √ |
| Chai et al. [48] (2021) | √ | √ | √ | × |
| Hsin Chang et al. [49] (2019) | √ | × | × | × |
| This paper | √ | √ | √ | √ |

## 3. Model and Assumptions

### 3.1. Model Description

This paper considers a closed-loop supply chain model composed of a single manufacturer and retailer [20]. The manufacturer produces two types of products: new and remanufactured products. The retailer is responsible for selling new products to the consumers. The manufacturer has opportunities to lower the variable cost of remanufacturing via process innovation, such as disassembly technology. Due to the high fixed costs of establishing recycling and remanufacturing operations; only the manufacturer is responsible for the remanufacturing of the product [31].

The manufacturer sells new products to retailers at the wholesale price of $w_n$, and then the retailer sells new products to consumers at the retail price of $p_n$. The manufacturer sells remanufactured products to consumers at the selling price of $p_r$; Government subsidies act on the manufacturer and has no direct relationship with the retailer and consumers. The product loses its life cycle after being used by consumers, and the manufacturer finally recovers the used products from consumers.

The benchmark model and the model with government subsidy in the CLSC are shown in Figure 1.

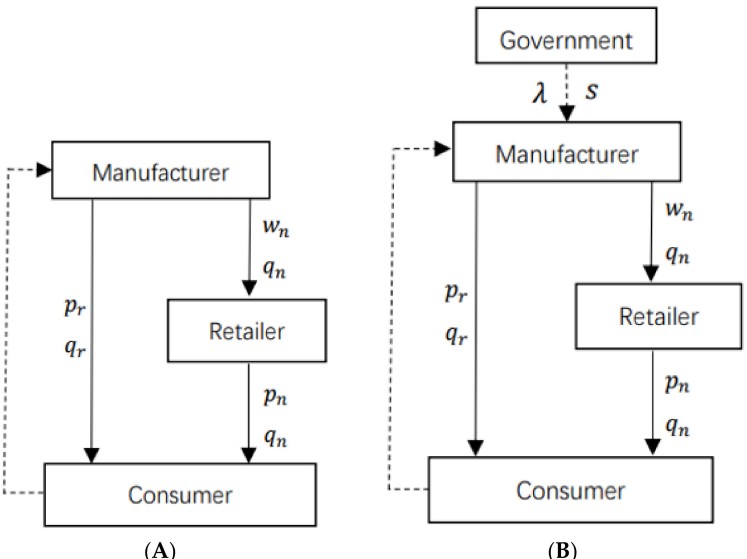

(A)  (B)

**Figure 1.** Two models within the closed-loop supply chain. Model (**A**) shows the benchmark model. Model (**B**) shows the model with government subsidy.

*3.2. Model Assumptions*

The purpose of this paper is to explore the impact of remanufacturing process innovation on the closed-loop supply chain remanufacturing from the perspective of government subsidy. In order to facilitate model calculation, the following hypotheses are proposed:

**Assumption 1.** *There is only one manufacturer and one retailer in the closed-loop supply chain. The manufacturer produces new and remanufactured products and reduces the production cost of remanufactured products through remanufacturing process innovation. There are remanufactured products in the market, and they are already in the supply chain cycle.*

**Assumption 2.** *New products and remanufactured products have no difference in function and quality, and they can be used for a period (a period in the life of the product). Consumers must make buying decisions to keep the market size constant in every period.*

**Assumption 3.** *All members are risk-neutral and completely rational [41].*

**Assumption 4.** *Assuming that the new product retail price is $p_n = Q - q_n - \alpha q_r$, and the selling price of manufactured products is $p_r = \alpha(Q - q_n - q_r)$ [21]. Q represents the market capacity. For the convenience of calculation, it is assumed that the market capacity Q = 1, that is, $p_n = 1 - q_n - \alpha q_r$, $p_r = \alpha(1 - q_n - q_r)$ [50].*

**Assumption 5.** *The remanufacturing process innovation level of remanufactured products stipulated by the government is $\lambda$, and its initial remanufacturing process innovation level is $\lambda_0$. The cost reduction caused by remanufacturing process innovation is $\lambda v$, and the production cost of remanufactured products is $c_r - \lambda v$, where v is the maximum cost reduction driven by remanufacturing process innovation [31]. Because of remanufacturing process innovation, manufacturers need to bear the costs of remanufacturing process innovation, $f = \frac{1}{2}\beta\lambda^2$, where $\beta$ is the remanufacturing process innovation cost coefficient [51]. Remanufacturing process innovation only affects the manufacturer, that is, only the manufacturer is engaged in the production of remanufactured products.*

**Assumption 6.** *The manufacturing cost per unit of new products is greater than that of remanufactured products, that is $c_n > c_r$.*

**Assumption 7.** *It is assumed that all new products can be recycled for remanufacturing after being used [52], so the cost of recycling is not considered, but the products can only be manufactured once. It is worth noting that recycling often turns the parts and components of waste products into raw materials. The products are low-level, and may consume a lot of energy and cause secondary environmental pollution. Remanufactured products consume less energy and are of higher grade.*

The related symbols are shown in Table 2.

**Table 2.** Model parameters.

| Symbol | Definition |
| --- | --- |
| $c_n$ | The unit production cost of new products |
| $c_r$ | The unit production cost of remanufactured products |
| $p_n$ | The retail price of new products |
| $p_r$ | The selling price of remanufactured products |
| $q_n$ | The quantity of new products |
| $q_r$ | The quantity of remanufactured products |
| $w$ | The wholesale price of new products |
| $v$ | Maximum cost reduction from remanufacturing process innovation |
| $\alpha$ | Consumers' acceptance of remanufactured products, $0 < \alpha < 1$ |
| $\beta$ | The remanufacturing process innovation cost coefficient |
| $\lambda$ | The level of remanufacturing process innovation set by the government, $0 < \lambda < 1$ |
| s | Government subsidy for the level of remanufacturing process innovation |
| $e_n$ | The environmental impact coefficient of the new products |
| $e_r$ | The environmental impact coefficient of the remanufactured products |
| ( )* | The optimal results |

## 4. The Models Development

Superscript "NI" and "I" respectively represent the closed-loop supply chain with remanufacturing process innovation but no government subsidy and the closed-loop supply chain with remanufacturing process innovation and government subsidy. The asterisk indicates the optimal decisions; Subscript $M$ and $R$ indicate the manufacturer and the retailer, respectively.

### 4.1. Model A: The Benchmark Models with Remanufacturing Process Innovation but No Government Subsidies

In this case, the overall profit of the closed-loop supply chain comes first, and both manufacturer and retailer have no conflict of interest. New and remanufactured products belong to different technological generations; that is, new products do not directly compete with remanufactured products [53]. The manufacturer sells remanufactured products directly to consumers at a price of $p_r$. Meanwhile, the retailer purchases new products from the manufacturer at a wholesale price of $w$ and sells the new products to consumers in $p_n$. But the government does not subsidize the closed-loop supply chain.

The profit-maximizing function of the closed-loop supply chain is

$$
\pi^{NI} = \underbrace{(p_n - c_n)q_n}_{\text{The new products' profit}} + \overbrace{(p_r - c_r + \lambda v)q_r}^{\text{The remanufactured products' profit}} - \underbrace{\frac{\beta \lambda^2}{2}}_{\text{The innovation costs}}. \tag{1}
$$

The profit-maximizing function of the manufacturer is

$$
\pi_M^{NI} = \underbrace{(w^{NI} - c_n)q_n}_{\text{The new products' profit}} + \overbrace{(p_r - c_r + \lambda v)q_r}^{\text{The remanufactured products' profit}} - \underbrace{\frac{\beta \lambda^2}{2}}_{\text{The innovation costs}}. \tag{2}
$$

The profit-maximizing function of the retailer is

$$
\pi_R^{NI} = \underbrace{(p_n - w^{NI})q_n}_{\text{The new products' profit}}. \tag{3}
$$

**Lemma 1.** *For the closed-loop supply chain without government subsidy, if $v^2 - 2\alpha\beta(1 - \alpha) < 0$, the optimal quantity of new products $q_n^{NI*}$, the optimal quantity of remanufactured products $q_r^{NI*}$, the optimal remanufacturing process innovation level of product $\lambda^{NI*}$ and the optimal wholesale price of new products $w^{NI*}$ can be obtained as follows:*

$$
q_n^{NI*} = \frac{v^2 - c_n v^2 - 2\alpha\beta + 2c_n\alpha\beta - c_r\alpha\beta + 2\alpha^2\beta}{2(v^2 - 2\alpha\beta + 2\alpha^2\beta)}
$$

$$
q_r^{NI*} = \frac{\beta c_r - c_n\alpha\beta}{v^2 - 2\alpha\beta + 2\alpha^2\beta}
$$

$$
\lambda^{NI*} = \frac{v(c_r - c_n\alpha)}{v^2 - 2\alpha\beta + 2\alpha^2\beta}
$$

$$
w^{NI*} = \frac{c_r\alpha\beta + c_n v^2 - 2c_n\alpha\beta + c_n\alpha^2\beta}{v^2 - 2\alpha\beta + 2\alpha^2\beta}
$$

Related proof is put in Appendix A.

Therefore, we can easily obtain the optimal retail price of new products $p_n^{NI*}$, the optimal selling price of remanufactured products $p_r^{NI*}$, the optimal profit of the manufacturer $\pi_M^{NI*}$, the optimal profit of the retailer $\pi_R^{NI*}$ and the optimal profit of closed-loop supply chain $\pi^{NI*}$.

**Corollary 1.** *(1) If* $c_n > \frac{(v^2 - 2\alpha^2\beta)c_r}{2\alpha(v^2 - \alpha\beta)}$, *there is* $\frac{\partial q_n^{NI*}}{\partial \alpha} > 0$, $\frac{\partial w^{NI*}}{\partial \alpha} < 0$; *If* $c_n > \frac{2\beta(2\alpha-1)c_r}{2\alpha^2\beta - v^2}$, *there is* $\frac{\partial q_r^{NI*}}{\partial \alpha} > 0$ *and* $\frac{\partial \lambda^{NI*}}{\partial \alpha} > 0$;

*(2) If* $c_n > \frac{c_r}{\alpha}$, *there is* $\frac{\partial q_n^{NI*}}{\partial \beta} > 0$, $\frac{\partial q_r^{NI*}}{\partial \beta} < 0$, $\frac{\partial \lambda^{NI*}}{\partial \beta} < 0$ *and* $\frac{\partial w^{NI*}}{\partial \beta} < 0$.

Related proof is put in Appendix B.

Corollary 1 indicates that consumers will increase their willingness to purchase remanufactured products and are more likely to make purchase behaviors with the increase of consumers' acceptance of remanufactured products. As their demand for remanufactured products increases, the manufacturer will produce more remanufactured products to satisfy consumers' demand. Meanwhile, the manufacturer will improve their remanufacturing process innovation level to achieve higher customer satisfaction. Furthermore, the manufacturer will stimulate the retailer to make a purchase decision by lowering the wholesale price of new products with the increase of consumers' acceptance and demand of remanufactured products, which in turn stimulates consumer demand for new products.

The greater the remanufacturing process innovation cost coefficient, the higher the costs of remanufacturing process innovation. And the manufacturer has to bear the costs generated in the process of remanufacturing process innovation, so the manufacturer will reduce the production quantity of remanufactured products in order to cut down the cost. Therefore, the manufacturer will lower the wholesale price of new products to the retailer in order to improve profits. At the same time, the manufacturer will inevitably reduce the level of remanufacturing process innovation to increase profits and reduce costs.

We can obtain the equilibrium results as summarized in Table 3.

**Table 3.** Equilibrium decisions and profits.

**(a)**: Model A

$$q_n^{NI*} = \frac{v^2 - c_n v^2 - 2\alpha\beta + 2c_n\alpha\beta - c_r\alpha\beta + 2\alpha^2\beta}{2(v^2 - 2\alpha\beta + 2\alpha^2\beta)}$$

$$q_r^{NI*} = \frac{\beta c_r - c_n\alpha\beta}{v^2 - 2\alpha\beta + 2\alpha^2\beta}$$

$$\lambda^{NI*} = \frac{v(c_r - c_n\alpha)}{v^2 - 2\alpha\beta + 2\alpha^2\beta}$$

$$w^{NI*} = \frac{c_r\alpha\beta + c_n v^2 - 2c_n\alpha\beta + c_n\alpha^2\beta}{v^2 - 2\alpha\beta + 2\alpha^2\beta}$$

$$p_n^{NI*} = \frac{1 + c_n}{2}$$

$$p_r^{NI*} = \frac{\alpha\left((1+c_n)v^2 - 2(1-\alpha)(c_r+\alpha)\beta\right)}{2(v^2 - 2\alpha\beta + 2\alpha^2\beta)}$$

$$\pi_M^{NI*} = \frac{(c_n\alpha - c_r)\beta\left(v^2(c_r + (c_n-2)\alpha) - 2\alpha(c_r - 2c_r\alpha + \alpha(c_n+2\alpha-2))\beta\right)}{2(v^2 - 2\alpha\beta + 2\alpha^2\beta)^2}$$

$$\pi_R^{NI*} = \frac{\left((c_n-1)v^2 - 2\alpha(c_n - c_r + \alpha - 1)\beta\right)^2}{4(v^2 - 2\alpha\beta + 2\alpha^2\beta)^2}$$

$$\pi^{NI*} = \frac{(c_n-1)^2 v^2 - 2\left(c_r^2 - 2c_n c_r\alpha + \left(1 + c_n^2 + 2c_n(\alpha-1) - \alpha\right)\alpha\right)\beta}{4(v^2 - 2\alpha\beta + 2\alpha^2\beta)}$$

**Table 3.** *Cont.*

**(b): Model B**

$$q_n^{I*} = \frac{v^2 - c_n v^2 + 2sv\alpha - 2\alpha\beta + 2c_n\alpha\beta - 2c_r\alpha\beta + 2\alpha^2\beta}{2(v^2 - 2\alpha\beta + 2\alpha^2\beta)}$$

$$q_r^{I*} = \frac{\beta c_r - c_n\alpha\beta - sv}{v^2 - 2\alpha\beta + 2\alpha^2\beta}$$

$$\lambda^{I*} = \frac{v(c_r - c_n\alpha) - 2s\alpha + 2s\alpha^2}{v^2 - 2\alpha\beta + 2\alpha^2\beta}$$

$$w^{I*} = \frac{c_r\alpha\beta + c_n v^2 - 2c_n\alpha\beta + c_n\alpha^2\beta - sv\alpha}{v^2 - 2\alpha\beta + 2\alpha^2\,\beta}$$

$$p_n^{I*} = \frac{1+c_n}{2}$$

$$p_r^{I*} = \frac{\alpha\left((1+c_n)v^2 - 2(1-\alpha)(c_r+\alpha)\beta + 2sv(1-\alpha)\right)}{2(v^2 - 2\alpha\beta + 2\alpha^2\beta)}$$

$$\pi_M^{I*} = \frac{1}{2(v^2 - 2\alpha\beta + 2\alpha^2\beta)^2}\left(-c_r^2\beta(v^2 + 2\alpha(2\alpha-1)\beta) + \alpha(-2sv^2(s+v) + ((c_n-2)c_n v^2 + \right.$$
$$4s^2(1-\alpha)^2 - 4sv(\alpha + c_n\alpha - 1))\alpha\beta - 2c_n\alpha^2(c_n + 2\alpha - 2)\beta^2) + 2c_r(sv^3 +$$
$$\left. v(v + s(4\alpha - 2))\alpha\beta + 2\alpha^2(\alpha + c_n\alpha - 1)\beta^2) - 2s(v^2 - 2\alpha\beta + 2\alpha^2\,\beta)^2\lambda_0\right.$$

$$\pi_R^{I*} = \frac{\left((c_n-1)v^2 - 2sv\alpha - 2\alpha(c_n - c_r + \alpha - 1)\beta\right)^2}{4(v^2 - 2\alpha\beta + 2\alpha^2\,\beta)^2}$$

$$\pi^{I*} = \frac{1}{4(v^2 - 2\alpha\beta + 2\alpha^2\beta)}\left(-4c_n sv\alpha - 2c_r^2\beta + 4c_r(sv + c_n\alpha\beta)\right.$$
$$+ v^2(1 - 2c_n + c_n^2 - 4s\lambda_0)$$
$$\left. + 2\alpha(2s^2(-1+\alpha) + (-1 + 2c_n - c_n^2 + \alpha - 2c_n\alpha)\beta - 4s(-1+\alpha)\beta\lambda_0)\right)$$

### 4.2. Model B: The Model with Remanufacturing Process Innovation and Government Subsidy

Under this circumstance, the government will give a different subsidy to the manufacturer who produces remanufactured products according to the intensity of remanufacturing process innovation. Here, $(\lambda - \lambda_0)s$ represents for the government subsidy to the manufacturer, where $\lambda_0$ is the initial remanufacturing process innovation level.

The profit-maximizing function of the closed-loop supply chain is

$$\pi^I = \underbrace{(p_n - c_n)q_n}_{\text{The new products' profit}} + \overbrace{(p_r - c_r + \lambda v)q_r}^{\text{The remanufactured products' profit}} + \underbrace{(\lambda - \lambda_0)s}_{\text{The government subsidies}} - \overbrace{\frac{\beta\lambda^2}{2}}^{\text{The innovation costs}}. \tag{4}$$

The profit-maximizing function of the manufacturer is

$$\pi_M^I = \underbrace{(w^I - c_n)q_n}_{\text{The new products' profit}} + \overbrace{(p_r - c_r + \lambda v)q_r}^{\text{The remanufactured products' profit}} + \underbrace{(\lambda - \lambda_0)s}_{\text{The government subsidies}} - \overbrace{\frac{\beta\lambda^2}{2}}^{\text{The innovation costs}}. \tag{5}$$

The profit-maximizing function of the retailer is

$$\pi_R^I = \underbrace{(p_n - w^I)q_n}_{\text{The new products' profit}}. \tag{6}$$

**Lemma 2.** *For the closed-loop supply chain with government subsidy, if $v^2 - 2\alpha\beta(1-\alpha) < 0$, the optimal quantity of new products $q_n^{I*}$, the optimal quantity of remanufactured products $q_r^{I*}$, the optimal remanufacturing process innovation level of product $\lambda^{I*}$ and the optimal wholesale price of new products $w^{I*}$ can be obtained as follows:*

$$q_n^{I*} = \frac{v^2 - c_n v^2 + 2sv\alpha - 2\alpha\beta + 2c_n\alpha\beta - 2c_r\alpha\beta + 2\alpha^2\beta}{2(v^2 - 2\alpha\beta + 2\alpha^2\beta)}$$

$$q_r^{I*} = \frac{\beta c_r - c_n\alpha\beta - sv}{v^2 - 2\alpha\beta + 2\alpha^2\beta}$$

$$\lambda^{I*} = \frac{v(c_r - c_n\alpha) - 2s\alpha + 2s\alpha^2}{v^2 - 2\alpha\beta + 2\alpha^2\beta}$$

$$w^{I*} = \frac{c_r\alpha\beta + c_n v^2 - 2c_n\alpha\beta + c_n\alpha^2\beta - sv\alpha}{v^2 - 2\alpha\beta + 2\alpha^2\ \beta}$$

Related proof is put in Appendix C.

Similarly, we can also obtain the others' equilibrium outcomes in Table 3.

**Corollary 2.** *(1) If $c_n > \frac{(c_r\beta - sv)(v^2 - 2\alpha^2\beta)}{2\alpha\beta(v^2 - \alpha\beta)}$, there is $\frac{\partial q_n^{I*}}{\partial\alpha} > 0$ and $\frac{\partial w^{I*}}{\partial\alpha} < 0$; if $c_n > \frac{2(1 - 2\alpha)(sv - c_r\beta)}{2\alpha^2\beta - v^2}$, there is $\frac{\partial q_r^{I*}}{\partial\alpha} > 0$ and $\frac{\partial\lambda^{I*}}{\partial\alpha} > 0$;*

*(2) If $c_n > \frac{c_r}{\alpha}$, there is $\frac{\partial q_n^{I*}}{\partial\beta} > 0$; if $c_n > \frac{c_r v - 2s\alpha(1 - \alpha)}{v\alpha}$, there is $\frac{\partial q_r^{I*}}{\partial\beta} < 0$, $\frac{\partial\lambda^{I*}}{\partial\beta} < 0$ and $\frac{\partial w^{I*}}{\partial\beta} < 0$.*

Related proof is put in Appendix D.

In this case, the government participates in the closed-loop supply chain. Moreover, the demand of consumers for remanufactured products will naturally increase with its acceptance. Moreover, because the government subsidizes the remanufacturing process innovation level of the manufacturer, it will indirectly stimulate the manufacturer to produce more remanufactured products and correspondingly improve the remanufacturing process innovation level to obtain higher government subsidy and customer satisfaction. The lower quantity for new products is caused by the lower wholesale price of new products as consumers accept remanufactured products more.

We can clearly see that remanufacturing process innovation costs increase with the remanufacturing process innovation cost coefficient increase. Although the government subsidizes the remanufacturing process innovation level, the income brought by the subsidy is less than the increase of the remanufacturing process innovation costs. The production cost of remanufactured products will increase instead, decreasing the production quantity. To reduce the production cost of remanufactured products, the manufacturer will inevitably choose to reduce the level of remanufacturing process innovation. Because of the higher production cost of remanufactured products, the manufacturer will increase the sales of their new products by reducing the wholesale price.

## 5. The Analysis and Comparison

### 5.1. The Analysis

Our analysis is based on the solutions of these models through backward induction. The equilibrium decisions and profits are obtained under this condition $v^2 - 2\alpha\beta(1 - \alpha) < 0$ and $c_n > c_r$ (which is $\frac{c_n}{c_r} > 1$).

**Corollary 3.** *When $\alpha > \frac{1}{2}$, the changes of the optimal decisions in model A are given by one of four possible structures, which is shown in Table 4.*

**Table 4.** The changes of the optimal decisions in Model A.

| | $\frac{q_n^{NI*}}{\alpha}$ | $\frac{q_r^{NI*}}{\alpha}$ | $\frac{\lambda^{NI*}}{\alpha}$ | $\frac{w^{NI*}}{\alpha}$ | $\frac{q_n^{NI*}}{\beta}$ | $\frac{q_r^{NI*}}{\beta}$ | $\frac{\lambda^{NI*}}{\beta}$ | $\frac{w^{NI*}}{\beta}$ |
|---|---|---|---|---|---|---|---|---|
| $\frac{c_n}{c_r} \leq \frac{2\beta(2\alpha-1)}{2\alpha^2\beta-v^2}$ | $-$ | $-$ | $-$ | $+$ | $-$ | $+$ | $+$ | $+$ |
| $\frac{2\beta(2\alpha-1)}{2\alpha^2\beta-v^2} < \frac{c_n}{c_r} \leq \frac{v^2-2\alpha^2\beta}{2\alpha(v^2-\alpha\beta)}$ | $-$ | $+$ | $+$ | $+$ | $-$ | $+$ | $+$ | $+$ |
| $\frac{v^2-2\alpha^2\beta}{2\alpha(v^2-\alpha\beta)} < \frac{c_n}{c_r} \leq \frac{1}{\alpha}$ | $+$ | $+$ | $+$ | $-$ | $-$ | $+$ | $+$ | $+$ |
| $\frac{c_n}{c_r} > \frac{1}{\alpha}$ | $+$ | $+$ | $+$ | $-$ | $+$ | $-$ | $-$ | $-$ |

Related proof is put in Appendix E.

According to Corollary 3, if $\alpha > \frac{1}{2}$, we can find that the changes of the optimal quantity of new products, the optimal quantity of remanufactured products, the optimal level of remanufacturing process innovation and the optimal wholesale price of new products is related to the proportion of new products of production cost and remanufactured products production cost. ()

**Corollary 4.** *When $\alpha > \frac{1}{2}$ and $\frac{2s\alpha^2}{v} < c_r < \frac{sv}{\beta}$, the changes of the optimal decisions in model B are given by one of five possible structures, which is shown in Table 5.*

**Table 5.** The changes of the optimal decisions in Model B.

| | $\frac{q_n^{I*}}{\alpha}$ | $\frac{q_r^{I*}}{\alpha}$ | $\frac{\lambda^{I*}}{\alpha}$ | $\frac{w^{I*}}{\alpha}$ | $\frac{q_n^{I*}}{\beta}$ | $\frac{q_r^{I*}}{\beta}$ | $\frac{\lambda^{I*}}{\beta}$ | $\frac{w^{I*}}{\beta}$ |
|---|---|---|---|---|---|---|---|---|
| $\frac{c_n}{c_r} \leq \frac{(c_r\beta-sv)(v^2-2\alpha^2\beta)}{2\alpha\beta(v^2-\alpha\beta)c_r}$ | $-$ | $-$ | $-$ | $+$ | $-$ | $+$ | $+$ | $+$ |
| $\frac{(c_r\beta-sv)(v^2-2\alpha^2\beta)}{2\alpha\beta(v^2-\alpha\beta)c_r} < \frac{c_n}{c_r} \leq \frac{2(1-2\alpha)(sv-c_r\beta)}{c_r(2\alpha^2\beta-v^2)}$ | $+$ | $-$ | $-$ | $-$ | $-$ | $+$ | $+$ | $+$ |
| $\frac{2(1-2\alpha)(sv-c_r\beta)}{c_r(2\alpha^2\beta-v^2)} < \frac{c_n}{c_r} \leq \frac{c_rv-2s\alpha(1-\alpha)}{c_rv\alpha}$ | $+$ | $+$ | $+$ | $-$ | $-$ | $+$ | $+$ | $+$ |
| $\frac{c_rv-2s\alpha(1-\alpha)}{c_rv\alpha} < \frac{c_n}{c_r} \leq \frac{1}{\alpha}$ | $+$ | $+$ | $+$ | $-$ | $-$ | $-$ | $-$ | $-$ |
| $\frac{c_n}{c_r} > \frac{1}{\alpha}$ | $+$ | $+$ | $+$ | $-$ | $+$ | $-$ | $-$ | $-$ |

Related proof is put in Appendix F.

Corollary 4 indicates that the changes of the optimal decisions with consumers' acceptance of remanufactured products and the remanufacturing process innovation cost coefficient is different when the proportion of new products production cost and remanufactured products production cost is not a fixed value, where $\alpha > \frac{1}{2}$ and $\frac{2s\alpha^2}{v} < c_r < \frac{sv}{\beta}$.

Based on the above analysis, whether it is the benchmark model with remanufacturing process innovation but no government subsidy, or the model with remanufacturing process innovation and government subsidy, the proportion of new products production cost and remanufactured products production cost will affect the changes of the optimal decisions. In terms of manufacturers' investment in remanufacturing, when the proportion of new product production cost and remanufactured product production cost is less than two but greater than one, if enterprises continue to improve the remanufacturing process innovation level in order to obtain more government subsidies, the economic benefits of remanufactured products will decrease. At this point, buying new products is more appropriate for consumers. When the cost of producing a new product is more than double the cost of producing a remanufactured product, it is advantageous for an enterprise to increase its investment in remanufacturing process innovation. Furthermore, enterprises can also get more government subsidies because of their higher level of remanufacturing process innovation. In terms of consumer acceptance of remanufactured products, due to the constraints of the proportion of new products production cost and remanufactured

products production cost, the manufacturer may not continue to improve remanufacturing process innovation level when consumers' acceptance of remanufactured products increases. However, when the remanufacturing process innovation cost increases, the manufacturer may reduce the remanufacturing cost by improving the remanufacturing process innovation level. This makes it possible to make up for the loss caused by the increase of process costs by increasing the marginal revenue of remanufactured products.

Furthermore, when the government gets involved in the closed-loop supply chain, the proportion of new products production cost and remanufactured products production cost is segmented and the changes of the optimal decisions are also different from the benchmark model. The government subsidy can stimulate the improvement of remanufacturing process innovation level. On the one hand, a high remanufacturing process innovation level will bring higher remanufacturing process innovation costs, but it will also get more remanufacturing cost savings. At this point, we should consider whether the marginal cost benefit brought by remanufacturing costs savings will be better than the loss brought by remanufacturing process innovation costs. On the other hand, although consumers have a higher acceptance of remanufactured products, the cost of remanufacturing process innovation brought by blindly improving the level of remanufacturing process innovation may also cause profit losses for the manufacturer. Therefore, enterprises should consider various factors when making decisions in practice instead of blindly improving the remanufacturing process innovation level to obtain government subsidies.

*5.2. The Comparison of Two Models*

This section compares the equilibrium decisions and profits obtained under the two different schemes.

**Proposition 1.** *(1) The manufacturer has higher remanufacturing process innovation level and wholesale price of new products under Model B; i.e., λI∗>λNI∗,wI∗>wNI∗.*
*(2) The manufacturer has lower selling price under Model B; i.e., prI∗<prNI∗.*
*(3) The retailer's new products retail price is equal in both models; i.e., pnI∗=pnNI∗*

Related proof is put in Appendix G.

Proposition 1 indicates that the manufacturer has an incentive to increase the level of remanufacturing process innovation when the government subsidizes the level of remanufacturing process innovation. Remanufacturing process innovation is a means for the manufacturer to reduce the production cost of remanufactured products, and government subsidy is equivalent to additional revenue. In order to obtain more subsidies, the manufacturer may try to improve the level of remanufacturing process innovation. However, the improvement of remanufacturing process innovation level will increase the cost of remanufacturing process innovation, which means that the manufacturer cannot blindly improve the remanufacturing process innovation level to obtain government subsidy. Meanwhile, we can see that the wholesale price of new products in Model B is higher than in Model A, but the selling price of remanufactured products is lower. When the government is involved in the closed-loop supply chain, the manufacturer shifts its efforts to manufacturing and selling remanufactured products. Therefore, the manufacturer will make a low-price strategy to survive in the market. Since there is no government subsidy for new products, the manufacturer will raise the wholesale price to improve the profits of new products. In addition, the retail price of new products is equal in the two models. Although the wholesale price of new products given by the manufacturer increases, the retailer does not increase the retail price of new products in order to retain market share.

**Proposition 2.** *Compared with the optimal decisions in the benchmark model with remanufacturing process innovation but no government subsidy, there is lower optimal quantity of new products and the higher quantity of remanufactured products under Model B; i.e., $q_n^{I*} < q_n^{NI*}$ and $q_r^{I*} > q_r^{NI*}$.*

Related proof is put in Appendix G.

The higher the remanufacturing process innovation level is, the more remanufactured product costs are reduced. Proposition 2 holds because the manufacturer is motivated to minimize remanufacturing costs to increase the production of remanufactured products by improving remanufacturing process innovation level when the government subsidizes the level of remanufacturing process innovation. The low-price strategy is also behind the increased demand for remanufactured products. The quantity for new products has declined because of the higher wholesale price and government subsidy for remanufacturing process innovation. A higher wholesale price means that sales of new products will decrease, and remanufacturing process innovation can bring the manufacturer costs reduction and additional subsidies. Hence, the manufacturer is more inclined to sell remanufactured products.

**Proposition 3.** *(1) If* $s < \frac{v\left(v^2 - 2\alpha\beta + 2\alpha^2\beta(1+c_n)\right)}{2\alpha\beta(\alpha-1)^2 - v^2}$, *there is* $\pi_M^{I*} < \pi_M^{NI*}$;

*(2) If* $s > \frac{2\alpha(1-c_n+c_r-\alpha)\beta+v^2(c_n-1)}{v\alpha}$, *there is* $\pi_R^{I*} > \pi_R^{NI*}$;

*(3) If* $s > \frac{\left(v^2-2\alpha\beta+2\alpha^2\beta\right)\lambda_0+v(c_n\alpha-c_r)}{\alpha(\alpha-1)}$, *there is* $\pi^{I*} > \pi^{NI*}$.

Related proof is put in Appendix H.

**Proposition 4.** *There is* $s_1 < s_2 < s_3$ *when* $0 < c_n - c_r - \lambda_0 v < 1 - \alpha < c_n$ *and* $-2\alpha^2\beta(1-\alpha) < v^2 - 2\alpha\beta + 2\alpha^2\beta < 0$, *where* $s_1 = \frac{v\left(v^2-2\alpha\beta+2\alpha^2\beta(1+c_n)\right)}{2\alpha\beta(\alpha-1)^2-v^2}$, $s_2 = \frac{2\alpha(1-c_n+c_r-\alpha)\beta+v^2(c_n-1)}{v\alpha}$ *and* $s_3 = \frac{\left(v^2-2\alpha\beta+2\alpha^2\beta\right)\lambda_0+v(c_n\alpha-c_r)}{\alpha(\alpha-1)}$.

Related proof is put in Appendix I.

Proposition 3 indicates that a government subsidy for remanufacturing process innovation does not necessarily improve the profits of supply chain members. The positive benefits brought by a government subsidy and remanufacturing costs saving by improving remanufacturing process innovation level are less than the increase of remanufacturing process innovation costs when $s < s_1 = \frac{v\left(v^2-2\alpha\beta+2\alpha^2\beta(1+c_n)\right)}{2\alpha\beta(\alpha-1)^2-v^2}$. Under these circumstances, the manufacturer may have no motivation to implement remanufacturing process innovation. As for whether the manufacturer can benefit from government subsidy, we will use the method of numerical analysis. And if $s > s_2 = \frac{2\alpha(1-c_n+c_r-\alpha)\beta+v^2(c_n-1)}{v\alpha}$, the retailer can benefit from government-subsidized remanufacturing process innovation. The government subsidy for remanufacturing process innovation can benefit the entire closed-loop supply chain when $s > s_3 = \frac{\left(v^2-2\alpha\beta+2\alpha^2\beta\right)\lambda_0+v(c_n\alpha-c_r)}{\alpha(\alpha-1)}$. Proposition 4 shows the relationship between $s_1$, $s_2$ and $s_3$. It provides a reference for how the government subsidy affects the change of remanufacturing process innovation level as well as under what circumstances reasonable government subsidies can bring positive profits to the whole supply chain or its members.

## 6. Numerical Analysis

In this section, we conduct several numerical analyses in order to illustrate our findings and obtain additional managerial insights.

### 6.1. Comparison of Two Models

Parameter values selected in this section are as follows: $c_n = 0.4$, $c_r = 0.1$, $v = 0.25$, $s = 0.01$, $\lambda_0 = 0.07$, $\alpha \epsilon (0.55, 0.6)$, $\beta \epsilon (0.25, 0.3)$.

Figure 2 shows the changes in the optimal quantity of new products, the optimal quantity of remanufactured products, the optimal remanufacturing process innovation level, and the optimal wholesale price in two models. In this case, the quantity of new products decreases with the consumers' acceptance of remanufactured products and increases with the remanufacturing process innovation cost coefficient in both models. The quantity of

remanufactured products, the remanufacturing process innovation level and the wholesale price increase with the consumers' acceptance of remanufactured products and decrease with the remanufacturing process innovation cost coefficient in both models.

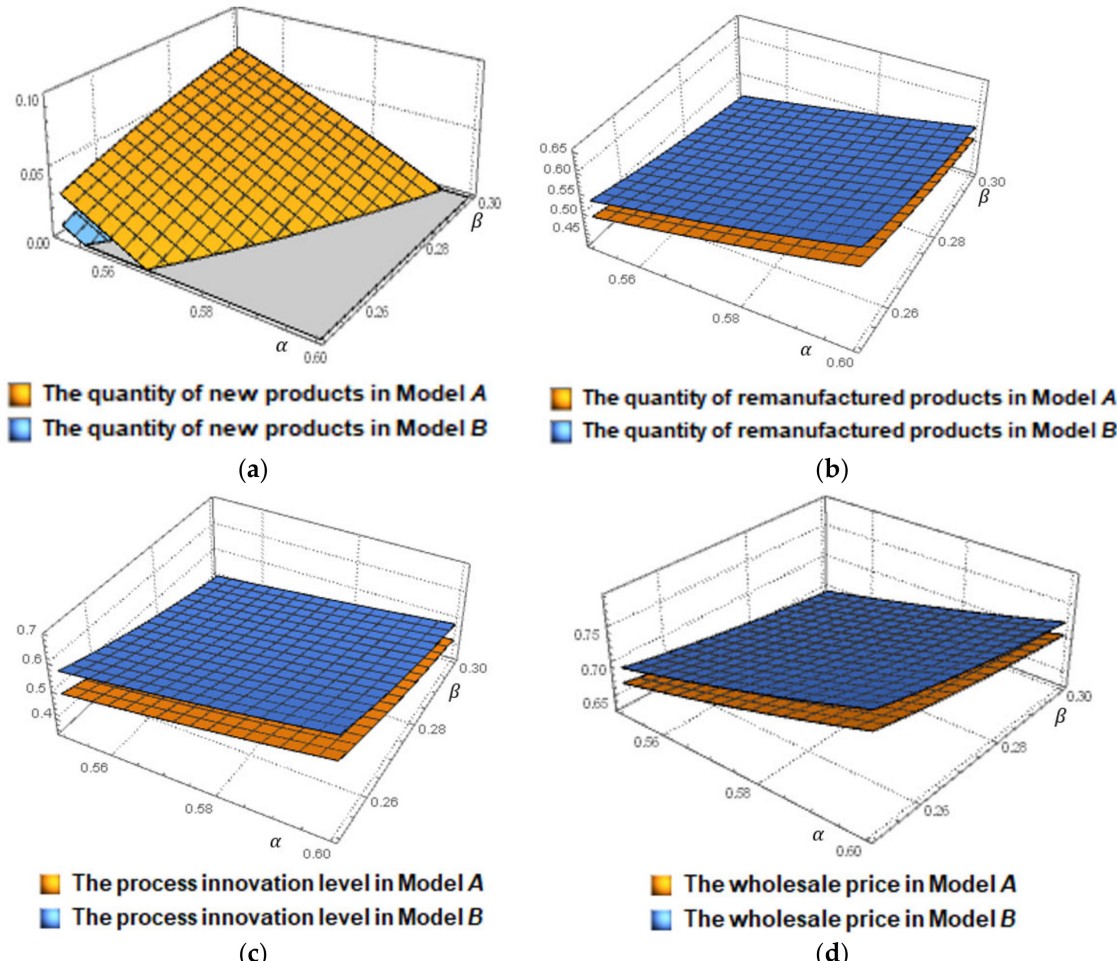

**Figure 2.** The equilibrium decisions of two models. (**a**–**d**) represents the quantity of new products, the quantity r of remanufactured products, the process innovation level, and the wholesale price in the two models, respectively.

On the one hand, the more consumers' acceptance of remanufactured products, the higher demand for remanufactured products. To achieve more remanufacturing cost savings, the manufacturer will be more motivated to improve the remanufacturing process innovation level. However, the increase in the demand for remanufactured products will reduce the demand for new products, so the manufacturer may increase the wholesale price to obtain more profit from new products.

On the other hand, with the increase of the remanufacturing process innovation cost coefficient, the costs of remanufacturing process innovation will increase. In order to cut down the remanufacturing process innovation cost, the manufacturer will inevitably reduce the level of remanufacturing process innovation. According to $c_r - \lambda v$, we can see that the production cost of remanufactured products is also influenced by remanufacturing process innovation level. Therefore, the sales of remanufactured products will decrease. In order to make more profits from new products, the manufacturer will also provide a low-price strategy to the retailer.

Compared with no government subsidy, there is a higher quantity of remanufactured products, remanufacturing process innovation level, and wholesale price of new products but a lower quantity of new products when the government subsidizes the level of remanufacturing process innovation. Government subsidies can stimulate the manufacturer to

remanufacture when the government is involved in a closed-loop supply chain. Therefore, compared with the situation without subsidy, although the production quantity of remanufactured products increases in both cases, it is evident that the production quantity of remanufactured products increases more under the case of government subsidy. Moreover, we can easily see that the higher the level of remanufacturing process innovation, the more government subsidy. New products have a smaller market share in subsidized markets, so the manufacturer needs to set higher wholesale prices to generate higher profits.

Figure 3 shows the changes in profits in two models. As consumers' acceptance of remanufactured products increases, the manufacturer's profit in the two models also increases, but the rate increases slowly. Meanwhile, the manufacturer's profit decreases slowly with the remanufacturing process innovation cost coefficient increase. As for the profit of the retailer, there are different situations. When the cost coefficient of remanufacturing process innovation is very small, the retailer's profit in model B increases with consumers' acceptance of remanufactured products. When the cost coefficient of remanufacturing process innovation is large, the retailer's profit in model B decreases first and then increases with consumers' acceptance of remanufactured products. However, in Model A, the retailer's profit continuously decreases first and then increases with consumers' acceptance of remanufactured products. When consumers have moderate acceptance of remanufactured products, retailers' profit in the two models will increase with the remanufacturing process innovation cost coefficient increase. However, the retailer's profit will decrease with the increase of remanufacturing process innovation cost coefficient in both models when consumers have a high acceptance of remanufactured products. Regardless of the models, the profit of the whole closed-loop supply chain increases with the increase of consumers' acceptance of remanufactured products, and decreases with the increase of remanufacturing process innovation cost coefficient.

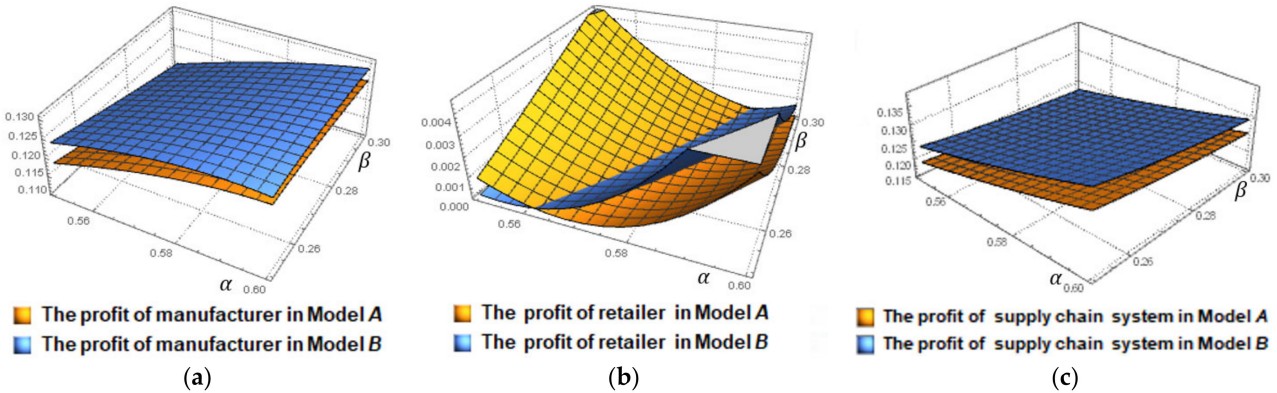

**Figure 3.** The profits of two models. (**a**–**c**) represents the profits of manufacturers, retailers, supply chain system in the two models, respectively.

Under this situation, the manufacturer's profits and closed-loop supply chain in Model B are always higher than in Model A. However, the retailer's profit in Model B is not always higher than in Model A, which is related to consumers' acceptance of remanufactured products and remanufacturing process innovation cost coefficient. To sum up, the retailer's profit may be hurt by government subsidies for remanufacturing process innovation, which sometimes reduces the retailer's profits when the government is involved in the supply chain. But the government subsidy for remanufacturing process innovation can benefit the manufacturer and the entire closed-loop supply chain system.

### 6.2. The Impact of Government Subsidy

We will explore whether different government subsidies have different effects on equilibrium decisions and profits. Parameter values selected in this section are as follows: $c_n = 0.4$, $c_r = 0.1$, $v = 0.25$, $\lambda_0 = 0.07$, $\alpha = 0.25$, $0.55$, $\beta = 0.25$, $0.55$.

Figure 4 shows four situations of $q_n$, $q_r$, $\lambda$ and w. We can easily see that the quantity of new products is always decreasing with the increase of government subsidy. And the quantity of remanufactured products, the level of remanufacturing process innovation, and the wholesale price of new products are always increasing with the increase of government subsidy. But they change at different rates. The quantity of remanufactured products is higher when $\alpha = 0.55$, $\beta = 0.25$. However, it also has the lower quantity of new products. If the remanufacturing process innovation cost coefficient is high, the government subsidy has little effect on $q_n$, $q_r$, $\lambda$ and w. The quantity of new products is higher than remanufactured products when $\alpha = 0.25$, $s < 0.03$. The quantity of new products is lower than remanufactured products when $\alpha = 0.55$.

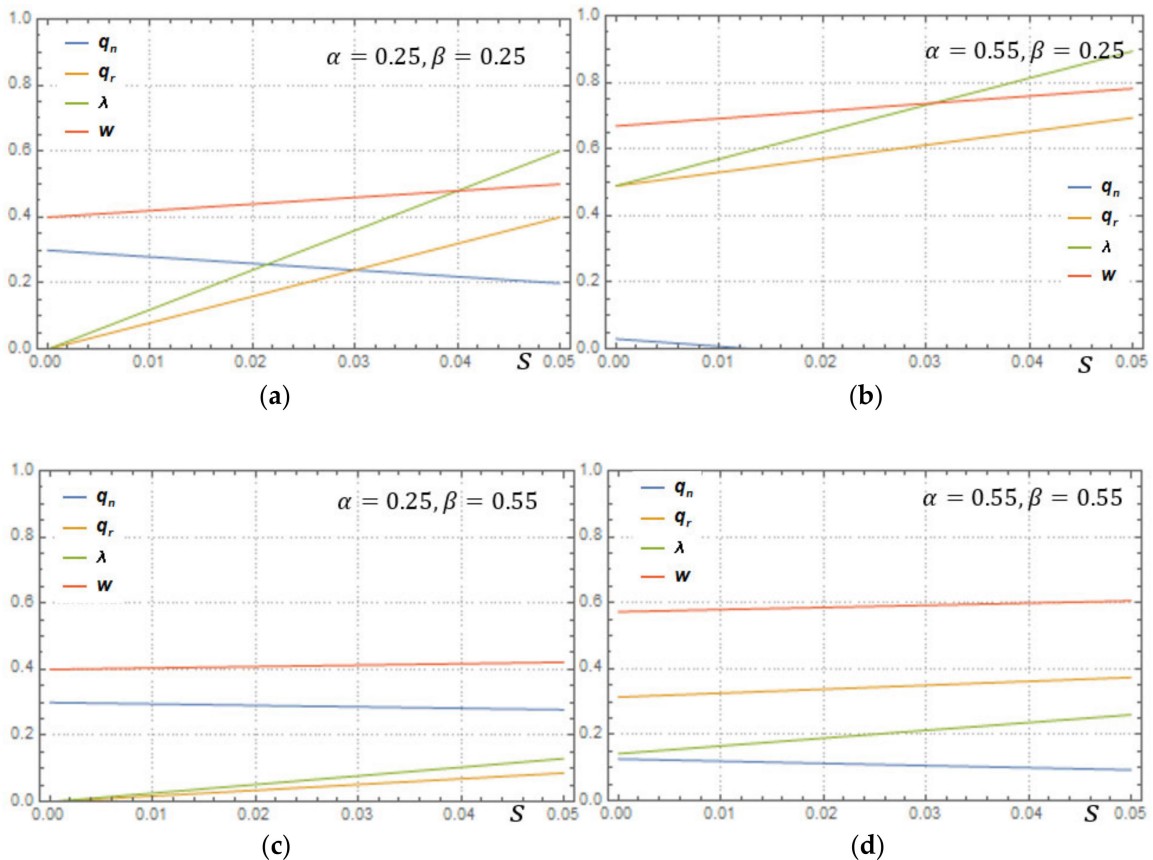

**Figure 4.** The impact of government subsidy on $q_n$, $q_r$, $\lambda$ and $w$. (**a**–**d**) represents four situations in which $\alpha$ and $\beta$ are in different values, which have been marked on the figure.

According to Figure 5, we can see that the improvement of government subsidy does not benefit everyone in the closed-loop supply chain. The costs of remanufacturing process innovation are still generated when the government subsidy for remanufacturing process innovation level is very small. However, if the government subsidy for remanufacturing process innovation level is less than the increase of remanufacturing process innovation costs, the profit for the manufacturer will be reduced. Therefore, the manufacturer may increase its profits by raising the wholesale price of the new products, which will hurt the retailer's profit. The manufacturer is less motivated by remanufacturing process innovation. When the government subsidy reaches a certain value, that is, the government subsidy for remanufacturing process innovation level is greater than or equal to the increase of remanufacturing process innovation costs, the manufacturer will implement remanufacturing process innovation and gradually improve the remanufacturing process innovation level to obtain a higher government subsidy. In this case, the profit of the closed-loop supply chain will also increase. In conclusion, there is a threshold $\bar{s}$, government subsidies are beneficial to the manufacturer, the retailer and the supply chain system when $s > \bar{s}$; when $s \leq \bar{s}$,

government subsidies can hurt the retailer's profits, and the retailer has no motivation to participate in the sale of new products.

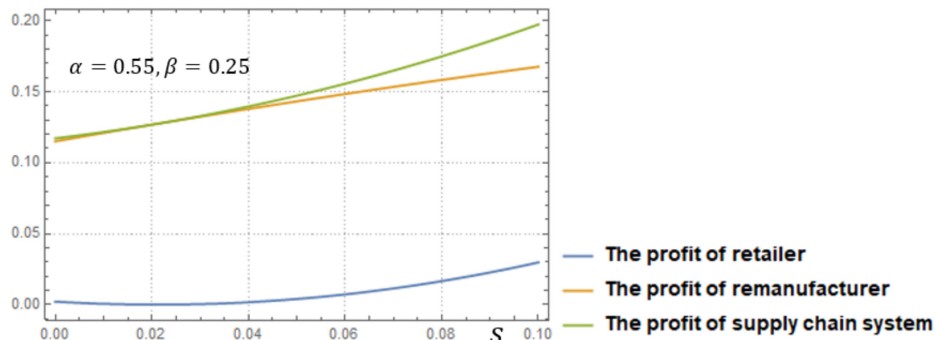

**Figure 5.** The impact of government subsidy on profits.

## 7. The Social Welfare

To explore the impact of government involvement on the whole society, we will explore the social welfare of the two models with numerical analysis. The social welfare consists of the profit of the manufacturer, consumer surplus, and environmental impact. We assume that $i \in \{A, B\}$.

The consumer surplus includes consumer willingness to purchase new products and remanufactured products. Therefore, the optimal consumer surplus is as follows [50]:

$$CS^A = \frac{1}{2}\left(1 - p_n^{NI*}\right)q_n^{NI*} + \frac{1}{2}\left(1 - p_r^{NI*}\right)q_r^{NI*} \tag{7}$$

$$CS^B = \frac{1}{2}\left(1 - p_n^{I*}\right)q_n^{I*} + \frac{1}{2}\left(1 - p_r^{I*}\right)q_r^{I*} \tag{8}$$

The first term represents the consumer surplus of new products, and the second term represents the consumer surplus of remanufactured products.

Meanwhile, the environmental impact refers to the environmental changes caused by the production of products, as well as the resulting social and economic effects. Moreover, the environmental impact is as follows:

$$E^A = e_n q_n^{NI*} + e_r q_r^{NI*} \tag{9}$$

$$E^B = e_n q_n^{I*} + e_r q_r^{I*} \tag{10}$$

We denote $e_j$ the environmental impact coefficient of the products, where $j \in \{n, r\}$. $e_n$ represents the environmental impact of new products and $e_r$ represents the environmental impact of remanufactured products ($e_n > e_r$). We can easily see that $E^A - E^B = \frac{-sv(e_n\alpha - e_r)}{v^2 - 2\alpha\beta + 2\alpha^2\beta}$ (where $v^2 - 2\alpha\beta + 2\alpha^2\beta < 0$).

Then, we can obtain the social welfare $SW^i$, which consists of the profit of the manufacturer, consumer surplus, and environmental impact:

$$SW^A = \pi_M^{NI*} + CS^A - E^A \tag{11}$$

$$SW^B = \pi_M^{I*} + CS^B - E^B. \tag{12}$$

The changes of the environmental impact are shown in Figures 6 and 7, which show the changes of social welfare in the two models. We set parameters: $e_n = 0.08$ and $e_r = 0.03$.

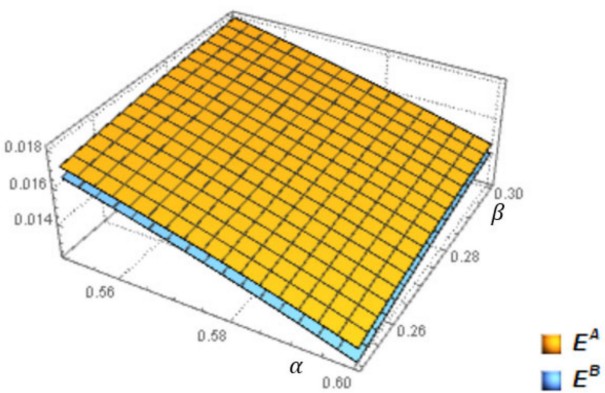

**Figure 6.** The environmental impact.

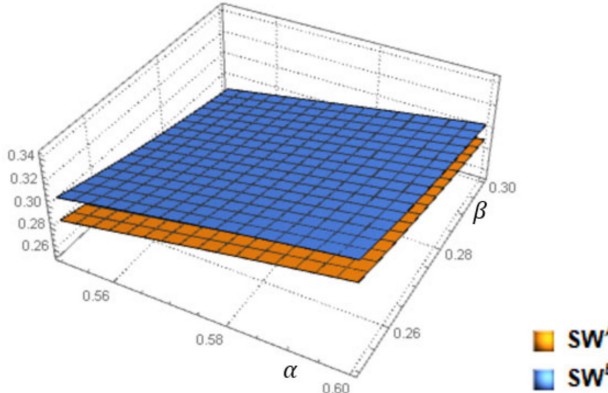

**Figure 7.** The social welfare.

According to Figure 6, we can also see $E^A > E^B$, which means that the products' production in Model A has a greater impact on the environment. Therefore, we can know that government subsidy for remanufacturing process innovation does reduce the environmental impact of production, which is beneficial for the environment. With the improvement of consumers' acceptance of remanufactured products, the impact of production on the environment is gradually decreasing. However, when the cost coefficient of remanufacturing process innovation increases, its impact on the environment will become larger. Figure 7 indicates that the social welfare gradually increases with consumers' acceptance of remanufactured products and decreases with remanufacturing process innovation cost coefficient. We can easily obtain $SW^A < SW^B$. In conclusion, if the government subsidizes the level of remanufacturing process innovation, it can improve the overall social welfare.

## 8. Conclusions and Future Research

Managing closed-loop supply chains with remanufacturing is a hot research topic because of its sustainable profile [31]. Remanufacturing has also become a way for enterprises to save resources and cut costs. Although remanufacturing costs can be reduced through remanufacturing process innovation, the relationship between remanufacturing cost reduction and remanufacturing process innovation cost increment becomes a problem that enterprises need to solve. Previous government subsidies were always in the form of direct subsidies. Therefore, we propose subsidizing remanufacturing process innovation according to the increase of remanufacturing process innovation level, in order to explore its influence on the whole closed-loop supply chain.

Our key findings are as follows:

(1) We obtain the equilibrium decisions and the optimal profits of the supply chain members in the two models and investigate their changes with the consumer acceptance of remanufactured products and remanufacturing process innovation level coefficient.

(2)　Whether it is the benchmark model with remanufacturing process innovation but no government subsidy or the model with remanufacturing process innovation and government subsidy, the optimal decisions of the two models vary with the proportion of new products' production cost and remanufactured products' production.

(3)　The government subsidy for remanufacturing process innovation does not always improve the profits of the manufacturer, the retailer and the supply chain system. The positive benefits brought by government subsidy and remanufacturing costs saving by improving remanufacturing process innovation level are less than the increase of remanufacturing process innovation costs when the government subsidy for the remanufacturing process innovation level is low. Under this situation, the manufacturer has no motivation to increase the remanufacturing process innovation level.

(4)　Through numerical analysis, it can be seen that government subsidy can benefit both the manufacturer and the whole supply chain system, but not always the retailer. The effect of government subsidy on the retailer is related to the relationship between consumers' acceptance of remanufactured products and remanufacturing process innovation cost coefficient. When consumers' acceptance of remanufactured products is high and the cost coefficient of remanufacturing process innovation is low, the government's subsidy for remanufacturing process innovation is beneficial to manufacturers, retailers, and supply chain systems. Moreover, there is a threshold $\bar{s}$ for government subsidies; government subsidies are beneficial to the manufacturer, the retailer and the supply chain system when $s > \bar{s}$. When $s \leq \bar{s}$, government subsidies can hurt the retailer's profits, and the retailer has no motivation to participate in the sale of new products.

(5)　When the government participates in the closed-loop supply chain and subsidizes the remanufacturing process innovation, the social welfare of the whole society is higher and the remanufacturing has less impact on the environment.

Our key findings have the following implication to the supply chain member. First, remanufacturing costs savings have been profitable for the manufacturer. No matter how small the cost reduction in the remanufacturing process, it can increase a manufacturer's profit margin. Then the manufacturer should try its best to lower the remanufacturing cost by the implementation of remanufacturing process innovation. However, the increased costs of remanufacturing process innovation sometimes can reduce the manufacturer's profits and make the manufacturer lose the initiative of remanufacturing process innovation. Secondly, the competitiveness of remanufactured products is weak when consumers' acceptance of remanufactured products is low. The retailer can take advantage of new products to increase its profits. When remanufactured products are competitive, the retailer should try to induce the manufacturer to lower wholesale prices; otherwise the profit will be damaged. Thirdly, from the perspective of the government, subsidizing remanufacturing process innovation is to encourage manufacturers to improve their remanufacturing process innovation level, and to improve the social welfare of the whole society. Government subsidies should be focused on small enterprises with limited innovation in existing processes in remanufacturing, because their processes or technologies may be outdated and capital is scarce.

This paper also gives some managerial insights. In practice, managers should not blindly improve the remanufacturing process innovation level to obtain higher government subsidies. Still, they should formulate a reasonable remanufacturing process innovation level according to consumers' acceptance of remanufactured products and the production cost of new and remanufactured products. Furthermore, enterprises tend to put their own profits first. Remanufacturing process innovation will not be attractive to enterprises when it cannot bring higher profits. For the government, the subsidies should be appropriately increased or decreased to encourage enterprises to remanufacture [54] or coordinate social welfare. Moreover, the government should focus on small enterprises with limited innovation of existing processes in remanufacturing, and realize gradual progress from point to surface.

This paper can be extended into several future directions. First, we only consider a single manufacturer and retailer in the supply chain, meaning that we ignore the power of different supply chain members. Future research can consider the impact of competition from third-party remanufacturers on the market. Secondly, this paper considers only one channel structure. In the future, different channel structures can be used to explore the impact of remanufacturing process innovation on closed-loop supply chain remanufacturing from the perspective of government subsidies. Thirdly, we don't take into account the quality of the recycled products used for remanufacturing. Therefore, we can consider the impact of recycled products quality on remanufacturing process innovation, so as to explore the dynamic impact of remanufacturing process innovation on closed-loop supply chain under the condition of used products' quality uncertainty.

**Author Contributions:** Conceptualization, K.L. and Q.L.; data curation, K.L. and H.Z.; formal analysis, K.L.; investigation, Q.L.; methodology, K.L.; project administration, K.L. and Q.L.; resources, Q.L. and H.Z.; software, K.L. and H.Z.; supervision, Q.L.; visualization, K.L. and H.Z.; writing—original draft preparation, K.L.; writing—review and editing, Q.L. and H.Z. All authors have read and agreed to the published version of the manuscript.

**Funding:** This research received no external funding.

**Institutional Review Board Statement:** Not applicable.

**Informed Consent Statement:** Not applicable.

**Conflicts of Interest:** The authors declare no conflict of interest.

**Appendix A**

**Proof of Lemma 1.** Hessian matrix is obtained from the profit function of closed-loop supply chain according to Formula (1):

$$H_1 = \begin{bmatrix} \frac{\partial^2 \pi^{NI}}{\partial q_n{}^2} & \frac{\partial^2 \pi^{NI}}{\partial q_n \partial q_r} & \frac{\partial^2 \pi^{NI}}{\partial q_n \partial \lambda} \\ \frac{\partial^2 \pi^{NI}}{\partial q_r \partial q_n} & \frac{\partial^2 \pi^{NI}}{\partial q_r{}^2} & \frac{\partial^2 \pi^{NI}}{\partial q_r \partial \lambda} \\ \frac{\partial^2 \pi^{NI}}{\partial \lambda \partial q_n} & \frac{\partial^2 \pi^{NI}}{\partial \lambda \partial q_r} & \frac{\partial^2 \pi^{NI}}{\partial \lambda^2} \end{bmatrix} = \begin{bmatrix} -2 & -2\alpha & 0 \\ -2\alpha & -2\alpha & v \\ 0 & v & -\beta \end{bmatrix}$$

where the first-order principal minor $|H_{11}| = -2 < 0$, second-order principal minor $|H_{12}| = 4\alpha(1 - \alpha) > 0$. And if $v^2 - 2\alpha\beta + 2\alpha^2\beta < 0$, the third-order principal minor $|H_{13}| < 0$, which the hessian matrix is negative definite, so we can obtain that $\pi^{NI}$ is a strict concave function about $q_n$, $q_r$, $\lambda$. Therefore, there is unique optimal solution for the quantity of new products, the quantity of remanufactured products and the level of remanufacturing process innovation when the closed-loop supply chain profit is maximized.

Combining $\frac{\partial \pi^{NI}}{\partial q_n} = 0$, $\frac{\partial \pi^{NI}}{\partial q_r} = 0$, $\frac{\partial \pi^{NI}}{\partial \lambda} = 0$, we can obtain the optimal results $q_n^{NI*}$, $q_r^{NI*}$, $\lambda^{NI*}$.

According to Formula (3), there is $\frac{\partial \pi_R^{NI}}{\partial q_n} = 1 - 2q_n - \alpha q_r - w$. Let the first partial derivatives of $\pi_R^{NI}$ with q$_n$ equal to 0. that is, $\frac{\partial \pi_R^{NI}}{\partial q_n} = 0$. We can easily verify that $w^{NI*} = 1 - \alpha q_r - 2q_n$. Then we substitute $q_n^{NI*}$ and $q_r^{NI*}$ into $w^{NI*}$, we can obtain that $w^{NI*} = \frac{c_r \alpha \beta + c_n v^2 - 2c_n \alpha \beta + c_n \alpha^2 \beta}{v^2 - 2\alpha \beta + 2\alpha^2 \beta}$.

Substituting $q_n^{NI*}$, $q_r^{NI*}$, $\lambda^{NI*}$ and $w^{NI*}$ into Formula (1), Formula (2) and Formula (3), we can also know the optimal profit of the closed-loop supply chain, the manufacturer and the retailer. □

## Appendix B

**Proof of Corollary 1.** According to $q_n^{NI*}$, $q_r^{NI*}$, $\lambda^{NI*}$ and $w^{NI*}$, there is $\frac{\partial q_n^{NI*}}{\partial \alpha}$ $= \frac{\beta\left(2c_n\alpha\left(v^2-\alpha\beta\right)-c_r\left(v^2-2\alpha^2\beta\right)\right)}{\left(v^2-2(1-\alpha)\alpha\beta\right)^2}$, $\frac{\partial q_r^{NI*}}{\partial \alpha}$ $= \frac{\beta\left(\left(2\alpha^2\beta-v^2\right)c_n-2\beta(2\alpha-1)c_r\right)}{\left(v^2-2(1-\alpha)\alpha\beta\right)^2}$, $\frac{\partial \lambda^{NI*}}{\partial \alpha}$ $= \frac{\beta\left(\left(2\alpha^2\beta-v^2\right)c_n-2\beta(2\alpha-1)c_r\right)}{\left(v^2-2\alpha\beta+2\alpha^2\beta\right)^2}$ and $\frac{\partial w^{NI*}}{\partial \alpha}$ $= \frac{\beta\left(2c_n\alpha\left(-v^2+\alpha\beta\right)+c_r\left(v^2-2\alpha^2\beta\right)\right)}{\left(v^2-2(1-\alpha)\alpha\beta\right)^2}$. So we can obtain that $\frac{\partial q_n^{NI*}}{\partial \alpha} > 0$ and $\frac{\partial w^{NI*}}{\partial \alpha} < 0$ when $c_n > \frac{c_r(v^2-2\alpha^2\beta)}{2\alpha(v^2-\alpha\beta)}$; if $c_n > \frac{2\beta(2\alpha-1)c_r}{2\alpha^2\beta-v^2}$, there is $\frac{\partial q_r^{NI*}}{\partial \alpha} > 0$ and $\frac{\partial \lambda^{NI*}}{\partial \alpha} > 0$.

Similarly, we can also know that $\frac{\partial q_n^{NI*}}{\partial \beta} = \frac{v^2\alpha(c_n\alpha-c_r)}{\left(v^2-2(1-\alpha)\alpha\beta\right)^2}$, $\frac{\partial q_r^{NI*}}{\partial \beta} = \frac{v^2(c_r-c_n\alpha)}{\left(v^2-2\alpha\beta+2\alpha^2\beta\right)^2}$, $\frac{\partial \lambda^{NI*}}{\partial \beta} = \frac{v(c_n\alpha-c_r)\left(2\alpha^2-2\alpha\right)}{\left(v^2-2\alpha\beta+2\alpha^2\beta\right)^2}$ and $\frac{\partial w^{NI*}}{\partial \beta} = \frac{v^2\alpha(c_r-c_n\alpha)}{\left(v^2-2(1-\alpha)\alpha\beta\right)^2}$. Therefore, if $c_n > \frac{c_r}{\alpha}$, there is $\frac{\partial q_n^{NI*}}{\partial \beta} > 0$, $\frac{\partial q_r^{NI*}}{\partial \beta} < 0$, $\frac{\partial \lambda^{NI*}}{\partial \beta} < 0$ and $\frac{\partial w^{NI*}}{\partial \beta} < 0$. □

## Appendix C

**Proof of Lemma 2.** The proof of Lemma 2 is similar to the proof of Lemma 1. □

## Appendix D

**Proof of Corollary 2.** The proof of Corollary 2 is similar to the proof of Corollary 1. □

## Appendix E

**Proof of Corollary 3.** According to Corollary 1, we assume that $x_1 = \frac{(v^2-2\alpha^2\beta)c_r}{2\alpha(v^2-\alpha\beta)}$, $x_2 = \frac{2\beta(2\alpha-1)c_r}{2\alpha^2\beta-v^2}$ and $x_3 = \frac{c_r}{\alpha}$. There is $x_1 - x_2 = \frac{c_r(v^2-2\alpha\beta+2\alpha^2\beta)^2}{2\alpha(v^2-\alpha\beta)(v^2-2\alpha^2\beta)}$. $v^2 - 2\alpha\beta + 2\alpha^2\beta < 0$, we can easily know that $c_r(v^2-2\alpha\beta+2\alpha^2\beta)^2 > 0$, which the numerator is greater than 0. As for $v^2 - \alpha\beta = v^2 - 2\alpha\beta + 2\alpha^2\beta + \alpha\beta - 2\alpha^2\beta = v^2 - 2\alpha\beta + 2\alpha^2\beta + \alpha\beta(1-2\alpha)$ and $v^2 - 2\alpha^2\beta = v^2 - 2\alpha\beta + 2\alpha^2\beta + 2\alpha\beta - 4\alpha^2\beta = v^2 - 2\alpha\beta + 2\alpha^2\beta + 2\alpha\beta(1-2\alpha)$, wo can obtain that the denominator is greater than 0 when $\alpha > \frac{1}{2}$. Therefore, $x_1 > x_2$.

Similarly, there is $x_1 - x_3 = -\frac{c_r(v^2-2\alpha\beta+2\alpha^2\beta)}{2\alpha(v^2-\alpha\beta)}$. We can also easily verify $x_1 < x_3$ when $\alpha > \frac{1}{2}$.
To sum up, there is $x_2 < x_1 < x_3$ when $\alpha > \frac{1}{2}$. □

## Appendix F

**Proof of Corollary 4.** Similarly, according to Corollary 2, we assume that $y_1 = \frac{(c_r\beta-sv)(v^2-2\alpha^2\beta)}{2\alpha\beta(v^2-\alpha\beta)}$, $y_2 = \frac{2(1-2\alpha)(sv-c_r\beta)}{2\alpha^2\beta-v^2}$, $y_3 = \frac{c_r}{\alpha}$ and $y_4 = \frac{c_rv-2s\alpha(1-\alpha)}{v\alpha}$. There is $y_2 - y_1 = \frac{(sv-c_r\beta)(v^2-2\alpha\beta+2\alpha^2\beta)^2}{2\alpha\beta(v^2-\alpha\beta)(v^2-2\alpha^2\beta)}$, we can easily know that $y_2 > y_1$ when $\alpha > \frac{1}{2}$ and $c_r < \frac{sv}{\beta}$. Moreover, there is $y_2 - y_3 = -\frac{2sv(1-2\alpha)+c_r(v^2-2\alpha\beta+2\alpha^2\beta)}{\alpha(v^2-2\alpha^2\beta)}$, $y_2 - y_4 = -\frac{(c_rv-2s\alpha^2)(v^2-2\alpha\beta+2\alpha^2\beta)}{\alpha(v^2-2\alpha^2\beta)}$ and $y_3 - y_4 = \frac{2s(1-\alpha)}{v}$. We can also obtain that $y_2 < y_3$ when $\alpha > \frac{1}{2}$, $y_2 < y_4$ when $\alpha > \frac{1}{2}$ and $c_r > \frac{2s\alpha^2}{v}$, $y_3 > y_4$.

We can easily verify $y_3 > y_4 > y_2 > y_1$ when $\alpha > \frac{1}{2}$ and $\frac{2s\alpha^2}{v} < c_r < \frac{sv}{\beta}$. □

## Appendix G

**Proof of Propositions 1 and 2.** According to Table 3, we can obtain that $q_n^{I*} - q_n^{NI*} = \frac{2sv\alpha}{2(v^2-2\alpha\beta+2\alpha^2\beta)} < 0$, $q_r^{I*} - q_r^{NI*} = -\frac{sv}{v^2-2\alpha\beta+2\alpha^2\beta} > 0$, $\lambda^{I*} - \lambda^{NI*} = -\frac{2s\alpha(1-\alpha)}{v^2-2\alpha\beta+2\alpha^2\beta} > 0$,

$w^{I*} - w^{NI*} = -\frac{2sv\alpha}{v^2 - 2\alpha\beta + 2\alpha^2\beta} > 0$, $p_n^{I*} - p_n^{NI*} = 0$ and $p_r^{I*} - p_r^{NI*} = \frac{sv(1-\alpha)\alpha}{v^2 - 2\alpha\beta + 2\alpha^2\beta} < 0$, where $v^2 - 2\alpha\beta + 2\alpha^2\beta < 0$. □

## Appendix H

**Proof of Proposition 3.** According to Table 3, we can see that $\pi_M^{I*} - \pi_M^{NI*} = \frac{1}{2(v^2-2\alpha\beta+2\alpha^2\beta)^2}s(c_r v(v^2 - 2\alpha\beta + 2\alpha^2\beta) + \alpha(-v^2(s+v) + 2\alpha(v + s(\alpha-1)^2 - (1+c_n)v\alpha)\beta) - (v^2 - 2\alpha\beta + 2\alpha^2\beta)^2\lambda_0)$, where $c_r v(v^2 - 2\alpha\beta + 2\alpha^2\beta)$ and $\left(-(v^2 - 2\alpha\beta + 2\alpha^2\beta)^2\lambda_0\right)$ is smaller than 0. So, there is $\pi_M^{I*} < \pi_M^{NI*}$ when $\alpha(-v^2(s+v) + 2\alpha(v + s(\alpha-1)^2 - (1+c_n)v\alpha)\beta) < 0$, that is, $s < \frac{v(v^2-2\alpha\beta+2\alpha^2\beta(1+c_n))}{2\alpha\beta(\alpha-1)^2 - v^2}$.

Similarly, there is $\pi_R^{I*} - \pi_R^{NI*} = \frac{sv\alpha(v(v-c_n v+s\alpha)+2\alpha(-1+c_n-c_r+\alpha)\beta)}{2(v^2-2\alpha\beta+2\alpha^2\beta)^2}$ and $\pi^{I*} - \pi^{NI*} = \frac{s(c_r v - \alpha(s+c_n v - s\alpha))}{v^2 - 2\alpha\beta + 2\alpha^2\beta} - s\lambda_0$. We can obtain $\pi_R^{I*} > \pi_R^{NI*}$ when $s > \frac{2\alpha(1-c_n+c_r-\alpha)\beta+v^2(c_n-1)}{v\alpha}$ and $\pi^{I*} > \pi^{NI*}$ when $s > \frac{(v^2-2\alpha\beta+2\alpha^2\beta)\lambda_0+v(c_n\alpha-c_r)}{\alpha(\alpha-1)}$. □

## Appendix I

**Proof of Proposition 4.** According to proposition 3, we can assume that $s_1 = \frac{v(v^2-2\alpha\beta+2\alpha^2\beta(1+c_n))}{2\alpha\beta(\alpha-1)^2-v^2}$, $s_2 = \frac{2\alpha(1-c_n+c_r-\alpha)\beta+v^2(c_n-1)}{v\alpha}$ and $s_3 = \frac{(v^2-2\alpha\beta+2\alpha^2\beta)\lambda_0+v(c_n\alpha-c_r)}{\alpha(\alpha-1)}$. There is $s_1 - s_2 = \frac{(v^2-2\alpha\beta+2\alpha^2\beta)^2(-1+\alpha+c_n)}{\alpha v(v^2-2(1-\alpha)^2\alpha\beta)} + \frac{2\beta c_r}{v}$, we can verify that $s_1 < s_2$ when $-2\alpha^2\beta(1-\alpha) < v^2 - 2\alpha\beta + 2\alpha^2\beta < 0$ and $c_n > 1 - \alpha$.

Similarly, there is $s_2 - s_3 = \frac{(v^2-2\alpha\beta+2\alpha^2\beta)(-1+\alpha+c_n-c_r+v\lambda_0)}{v((1-\alpha)\alpha)}$. If $0 < c_n - c_r - \lambda_0 v < 1 - \alpha$, we can know that $s_2 < s_3$.

Having said all of above, we can obtain that $s_1 < s_2 < s_3$ when $0 < c_n - c_r - \lambda_0 v < 1 - \alpha < c_n$ and $-2\alpha^2\beta(1-\alpha) < v^2 - 2\alpha\beta + 2\alpha^2\beta$. □

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
