# Peer review of "Analysis of the Impact of Remanufacturing Process Innovation on Closed-Loop Supply Chain from the Perspective of Government Subsidy"

_sustainability, doi:10.3390/su141811333_

Round 1

Reviewer 1 Report (Previous Reviewer 1)

There are no further questions for me.

Author Response

Thank you very much for your help.

Reviewer 2 Report (Previous Reviewer 2)

Thank you to the Authors for referring to all my comments, questions and doubts, and for comprehensive answers and so many corrections. I wish your model will be effectively used in practice.

Author Response

Thank you for your help with our manuscript.

Reviewer 3 Report (New Reviewer)

Please see the attached comments.

Author Response

Many thanks for your valuable comments and suggestions for this article. We have carefully addressed the comments and suggestions in the revised version. 

We provide an attachment for your convenience.

Reviewer 4 Report (New Reviewer)

  1. This manuscript presents a closed-loop supply chain model considering remanufacturing, process innovation and government subsidy. The proposed model maximizes the profit of the supply chain to obtain the optimal quantity of remanufactured products. Some of the corrections are as follows.

    1. The abstract should be revised. It should be content-specific.
    2. In the introduction section, different model elements are introduced sequentially, and there is no coherence between different paragraphs. The role of the introduction is to highlight the importance of the research and pose some research questions which cannot be seen in the introduction section of this paper.
    3. The literature section is not well-structured, and there is no coherence between different paragraphs. Also, the literature is not comprehensive. 
    4. The language of the paper needs a complete revision. There are several typos and errors which should be corrected.
    5. Revise Table 2. The symbol of the parameters does not visualize correctly.
    6. Revise the paper, including managerial insights of this problem and the novelty of this paper compared to the existing literature.
    7. The results are not linked as properly with implications and conclusions.
    8. The conclusion section is weak, and the readability should be increased.

Author Response

We would like to thank you for your careful reading, helpful comments, and constructive suggestions, which has significantly improved the presentation of our manuscript.

We provide an attachment for your convenience.

Round 2

Reviewer 3 Report (New Reviewer)

I appreciate the authors made their best effort to provide further elaborations on the matters addressed in the revision process.  

Reviewer 4 Report (New Reviewer)

The authors have improved the manuscript and the suggested corrections were included in the second version.

This manuscript is a resubmission of an earlier submission. The following is a list of the peer review reports and author responses from that submission.

Round 1

Reviewer 1 Report

This paper investigates the impact of government subsidy on the process innovation of the remanufacturing closed-loop supply chain. I have some concerns in the following.

1) The motivation for this study is unclear. As the authors noted in their paper, process innovation is distinct from product innovation and technology innovation. The process innovation refers to innovations in operational management. It seems that the government is not willing to provide incentives for this kind of innovation. In general, governments are willing to provide subsidies for research or technology development.

2) The authors should further summarize their contributions to the existing literature.

3) The manufacturer sells new products through the retailer and remanufactured products directly to consumers. Why is this assumption? The manufacturer may also be able to sell new products directly to consumers if it is able to sell remanufactured products directly to consumers.

4) The authors assume that the new product and remanufactured product have no differences in function and quality. What does alpha in the reverse demand function mean?

5) The supply chain model is quite simple and the contribution is insignificant.

Reviewer 2 Report

The problem described in the article seems to me to be very interesting both from a scientific and a practical point of view.

We know from our everyday lives that it is often not profitable to repair devices and it is better to buy new ones. This has an impact on the natural environment. So, as I understand it, it implies the need for state support? Such an assumption was, as I understand it, adopted in advance by the authors?

However, as I understand it, the authors suggest that the solution is innovations that are supposed to reduce the cost of repairs. Only then what is the purpose of subsidization? For research on these innovations? I think that such justification should appear at the beginning of the article.

If the state supports the implementation of innovations, from the social point of view, these costs are still high. Only, I understand that the costs of developing a given technology („high fixed costs”) and its implementation are spread over a larger number of companies? Perhaps it is worth discussing this issue, at least briefly, what is the role of the state in the development of innovation.

What this sentence from the introduction mean:

„There are few studies that subsidize remanufacturing based on the level of improvement in process innovation.”

May be:

„studies that adress the problem of …”

I think that the authors should clearly define the purpose of the research - for example, "the impact of models of subsidizing innovation on the profits of companies"?

What this sentence means:

„And it is no significant if the remanufacturing renders the product not cost-effective.”

Why „it is no significant”?

So apart from Reimann et al. (2019) and Chai et al. (2021), this this topic has not been discussed before by other authors? Please explain to the layman what your new approach to the problem is. Because when I read the last (most important) paragraph of the literature review, I do not understand it.

I am also not able to assess the correctness of the formulas used for the simulations and I do not know what data the authors used to carry out these simulations.

All the more, the authors should clearly explain the conclusions resulting from the conducted research. It seemed to me that they were reading the beginning of the work that the point was to show the conditions under which it is necessary to subsidize innovation by the government, or when it becomes profitable for enterprises.

I am not sure if such statements can be made:

„In order to obtain more subsidies, the manufacturer will undoubtedly improve the level of process innovation.”

Why „undoubtedly”? We cannot be sure what decision the entrepreneur will make, because it does not result from mathematical formulas, but from his subjective assessment.

It seems to me that the essence of the problem is contained in 4 points. conclusions:

„ …the proportion of new products production cost and remanufactured products production cost will affect the changes of the optimal decisions.”

So, for example, with what share of fixed costs it is necessary to provide support from the state, what this level should be, etc.

There is a regularity that I notice in people dealing with the issues of modeling, simulation, and operations research - that first the essence of the problem is presented in a very general way, and then there is a sudden jump to very complex mathematical formulas, completely incomprehensible to a person not specialized in such issues. I would recommend the authors to first present a simple mathematical formula with a clearly defined optimization criterion.

Without it, I cannot judge the quality of the article.

Reviewer 3 Report

The paper studies the problem of process innovation from a perspective of closed-loop systems. This is interesting. The study presents some concepts and a numerical evaluation using two models: with and without government subsidies. The analysis is theoretical, by considering a very simplified, yet complex to mathematically evaluate, supply chain. This approach does not allow actual evaluation of government policies for actual decision making. I suggest to consider a real case study from an organization for which these problems are of relevance and carry out empirical research to actually assess the impact of government policies. 

Some conclusions are contradictory, and need to be better explained in detail. For example, authors write in the abstract: "The government subsidy for process innovation does not necessarily improve the profits of supply chain members. When government subsidy is appropriate, the government subsidy can benefit both the manufacturer and the whole supply chain, but not always the retailer. ". This type of sentences must be developed in full detail within the text and written in a different manner in the abstract to avoid confusions.

Some parts of the paper must be reorganized, in terms of formatting or in terms of content. For example, the introduction is too long, and includes  the research questions. I suggest to split this section, by shortening its contents focusing only on the general context of the study, presenting the objective of the paper and the structure of the manuscript. This introductory section is followed by the review of literature. Then, a third section can present the problem under study, the research questions and the research methodology. Section 3 can be this last section.

In regard to formatting, Section 3 currently starts with a figure, while it is recommended to start with the text and then embed the figure in an appropriate place.